# Building a Prevention System: Infrastructure to Strengthen Health Promotion Outcomes

**DOI:** 10.3390/ijerph18041618

**Published:** 2021-02-08

**Authors:** Monica Bensberg, Andrew Joyce, Erin Wilson

**Affiliations:** Centre for Social Impact, Faculty of Business and Law, Swinburne University of Technology, P.O. Box 218, Mail H25, Hawthorn, VIC 3122, Australia; ajoyce@swin.edu.au (A.J.); ewilson@swin.edu.au (E.W.)

**Keywords:** systems thinking, health promotion, prevention system

## Abstract

Prevention systems improve the performance of health promotion interventions. This research describes the establishment of the Australian state government initiative, Healthy Together Victoria’s (HTV) macro infrastructure for the delivery of large-scale prevention interventions. Methods: This paper reports on findings of 31 semi-structured interviews about participants’ understanding of systems thinking and their reflections of the strengths and weaknesses of the HTV prevention system. A chronic disease prevention framework informed the coding that was used to create a causal loop diagram and a core feedback loop to illustrate the results. Results: Findings highlighted that HTV created a highly connected prevention system that included a sizeable workforce, significant funding and supportive leadership. Operating guidelines, additional professional development and real-time evaluation were significant gaps, which hindered systems practice. For inexperienced systems thinkers, these limitations encouraged them to implement programs, rather than interact with the seemingly ambiguous systems methods. Conclusions: HTV was an innovative attempt to strengthen health promotion infrastructure, creating a common language and shared understanding of prevention system requirements. However, the model was inadequate for HTV to achieve population-level reductions in chronic disease as system oversight was missing, as was an intervention delivery focus. Clarity was needed to define the systems practice that HTV was seeking to achieve. Importantly, the HTV prevention system needed to be understood as complex and adaptive, and not prioritized as individual parts.

## 1. Introduction

Systems thinking is an approach for solving complex problems, that emphasises looking at the whole rather than the isolated parts, and highlighting the relationships between the parts, their causal linkages and feedback loops [1,2]. The application of systems thinking to prevention is experienced at two levels [3]. The first is the micro level where health promoters are working *in* the system, achieving change with the local application of systems practices to tackle public health challenges [4,5]. The context in which this community-based implementation exists is nested within the second broader or macro level where organisations are working *on* the system, to develop a structure to coordinate effective whole-of-population strategies [6,7]. This infrastructure is referred to as a prevention system.

Prevention systems improve the performance of health promotion interventions [8]. In Australia, in 1996, reductions in mortality from HIV/AIDS, smoking and road traffic injury were attributed to a national structure, which was recently deemed as having a crucial role in preventing the spread of COVID-19 [9,10]. Success was attributed to the entire system’s performance, not to any single individual components, for instance, good strategy or generous funding individually do not necessarily lead to the desired outcomes [6,11]. A prevention system includes the people, processes, activities, settings and structures, as well as the changing relationships between them, which work together to improve the health of a community [12]. Their core functions usually consist of policy, monitoring and surveillance, financing, research and evaluation, workforce, and program delivery [9] that together assist to recognise a system’s strengths and diagnose weak links to improve them [13].

There is limited research about the use of systems sciences pertaining to the macro environment in public health [14,15]. This paper uniquely describes the establishment of Healthy Together Victoria’s (HTV) infrastructure for a systems approach to prevention [16]. 

### Theoretical Background

Since the early 2000s there have been ongoing requests in the literature calling for the use of systems thinking in health promotion, mainly to fill a void in the theory and practice regarding effectively tackling complex public health problems [5,17,18].

Systems thinking is drawn from the overlapping sciences of systems theory and complexity theory [19], where system dynamics and soft systems methodology are considered to be the relevant approaches to health promotion [5,20,21] that are currently being used in community-based prevention [22,23,24,25,26]. These methods shift the focus of health promotion activities away from a reductionist perspective to a holistic one [27], recognising that the essence of structure is not the parts themselves, but the relationships between the parts, their causal connections [2]. This is a fundamental shift in thinking for most health promotion professionals moving from objectives-based planning and considering ‘Where do I implement a project?’, to problem-solving processes that view the complexity of an issue by asking ‘What needs to change to improve the situation?’ [28]. Systems thinking for prevention requires significant expertise to map systems and locate leverage points to alter their behaviour [29]. At macro and micro levels systems practice requires the reflecting, learning and adapting cycles of emergent strategy to facilitate system interactions via convening stakeholders, engaging communities, connecting agencies, coordinating assets, testing interventions, building consensus and aligning activities [30,31,32]. It involves taking risks in dynamic environments and continuously learning from them to prevent obesity, family violence, diabetes and other chronic problems [33].

## 2. Materials and Methods

### 2.1. Study Setting

Within Australia, the Victorian Government identified the need for a new prevention system, as the existing one, known as Integrated Health Promotion (IHP), was ineffective [34]. It was located in community health services and was criticized for delivering fragmented and short-term projects [16,35]. IHP was to be deliberately disrupted by Healthy Together Victoria (HTV), a large-scale intervention that adopted systems thinking to reduce obesity and preventable chronic disease from 2012 to 2016 [36]. HTV aimed to improve the effectiveness of health promotion in the state of Victoria [16]. The theory of change was a complex whole of system approach that required the activation of community-level organisations via multifaceted interventions to improve physical inactivity, poor diet, smoking and harmful alcohol use [29,37]. HTV focused on multiple environmental settings and the underlying structure of a problem, including its interrelated causes, to facilitate individual and community change [14,38]. HTV was testing systems thinking interventions at micro- and macro-levels that, together, were believed to be capable of improving population-level health outcomes [7,39].

#### 2.1.1. Local Prevention Platform for Micro-Level Interventions

Twelve Healthy Together Communities (HTC) were established in local governments, with teams of 10–15 people in designated roles, such as senior coordinators, evaluators and health promotion officers [40]. This workforce had developed an understanding of complexity theory. Their work was adaptive within the local context, consistent with emergent practice [35]. They were testing systems thinking tools and skills to achieve health-promoting goals [41,42]. These previously inexperienced systems thinkers were grasping the new prevention agenda [43].

HTC also implemented HTV’s programs, designed at the macro-level for state-wide delivery, such as the Achievement Program (creating healthy kindergartens, schools and workplaces) and the Healthy Living Programs (evidence-based behaviour change interventions) [44,45].

#### 2.1.2. State-Wide Prevention Platform for Macro-Level Interventions

More broadly, the design of the HTV prevention system provided the architecture for a state-wide health promotion infrastructure made up of five building blocks (blocks):
LEADERSHIP: Building champions, from local community members to political leaders, to drive health-promoting change.KNOWLEDGE and data: Capturing information on HTV’s progress with new types of monitoring and evaluation. The Centre of Excellence in Intervention and Prevention Science (CEIPS) was responsible for program evaluation.RESOURCES: In excess of 120 million dollars was provided by state and federal governments to fund HTV.WORKFORCE: Approximately 130 people were employed in the prevention workforce, predominately in the HTC. They were described as ‘a systems thinking and acting workforce to combat chronic disease’.RELATIONSHIPS: Building relationships with prevention partners, across sectors and industries, for engagement in collective action [38,46].

Each block is a subsystem that performs a function towards health promotion outcomes, while driving change through its connections to other blocks, where a change in one part causes change in another. As was understood within HTV, ‘the blocks alone do not constitute a system, it is the interactions among them—that convert the blocks into a system’ [47]. 

Previous research on adopting systems approaches has examined the micro-level of community-based practice in HTC [5,41,42,43] and other papers explore various aspects of HTV [14,16,35,41,43,44]. This paper is the first to appreciate the macro-level architecture of system design, examining how the HTV prevention system was constructed and its strengths and limitations.

### 2.2. Study Design

During HTV’s final year of implementation, in 2016, 31 primary interviews were conducted with: six participants from the Victorian State Government Department of Health and Human Services (DHHS) including senior and policy advisors; three participants from CEIPS, including senior staff and an officer; and 22 participants from HTC including eight coordinators, four evaluators, six health promotion officers and four team leaders (see Table 1). One person declined to participate, and another person did not reply to the invitation to partake in the research. The participants had held a role in influencing or implementing HTC’s systems approach to prevention for at least ten-months. The initial participants were identified using purposive sampling and subsequent participants were nominated via snowball recruitment [48,49]. They were contacted by telephone and emailed additional information. An interview guide was prepared and pre-tested with questions focusing on participants’ understanding of systems thinking and how it had been applied in HTV. The interviewer was unknown to the participants. Interviews were mostly conducted in the participants’ workplaces. Interviews ranged in duration from 23 to 120 min (average 60 min) and were recorded and transcribed verbatim. The participant’s name was not recorded or included in the transcript. Other factors that may have identified a participant were removed from the transcript. Two participants took up the offer to audit their transcript. After each interview, the interviewer documented key reflections. Data was analysed in NVivo^TM^ software (version 12). This research was conducted independently and external to the HTV evaluation.

Deductive thematic analysis enabled patterns to emerge from the data concerning participants’ lived experiences regarding systems thinking for prevention [33]. Braun and Clarke’s (2006) six-phase framework for undertaking thematic analysis was followed: reading and re-reading the data to be familiar with it, generating codes, searching for themes to identify meaning, refining themes, and determining the story of each theme and documenting them [50]. The themes were considered beside the interviewer’s key reflections. The analysis was undertaken by one researcher and discussed with a second to reach consensus on coding and themes. A third researcher had been invited to resolve any analysis discrepancies, but this was not needed.

Based on a systematic review, an interpretative framework by Baugh Littlejohns and Wilson (2019), ‘Strengthening systems for chronic disease prevention’, informed the coding [51]. It identified the parts needed for an effective chronic disease prevention system. While other frameworks are available [52,53], this one was selected as it is contemporary and takes a complexity-informed approach. Table 2 compares the blocks of the HTV prevention system with the parts of the interpretative framework. This research examines how the parts of Baugh Littlejohns and Wilson’s framework may have been present (or absent) in the HTV prevention system.

A causal loop diagram (CLD) and a core feedback loop (produced in Vensim^TM^ software) summarize the findings, using Kim and Andersen’s (2012) coding method for system dynamics to generate maps using qualitative text [54]. This involved analyzing the themes for explicit or implied parts of the prevention system and the direction of the causal linkages between them to develop the CLD and core feedback loop [23]. For continuity in the findings, the headings of the parts are based on those in the HTV building block and Baugh Littlejohns and Wilson’s framework.

## 3. Results

### 3.1. Interview Findings

The results are described using themes from the interpretative framework (see Table 2) as subheadings, initially discussing the individual parts of the prevention system and then the whole.

#### 3.1.1. The Parts

As discussed above, the HTV prevention system comprised five blocks. It will be shown that certain blocks received adequate attention, while others received less than they probably should have.

a. Leadership

Leadership for systems thinking was well established for HTV as, per the literature, systems leadership required political will and multiple levels of leaders’ support to galvanize a public health movement. Hierarchical leadership was visible with respect to: the Minister for Health; the responsible State Government Department Senior Advisor; the local government mayors and executive officers; government representatives on the HTC governance groups; and the HTC Coordinators. Regarding the Minister’s involvement, a participant recalled that:
“The engagement of the Minister! To have them come out to every HTC site—it was huge support.”(HTC Coordinator, P8)


Regarding the mayors and executive officers, another participant commented that:
“That leadership was great. … Having a CEO on my governance group and to chair the internal health and well-being committee, and then started the work around a health promoting workplace. He ended up being a great advocate for Healthy Together. He also talked to MPs and spent time building this leadership.”(HTC Coordinator, P7)


Regarding the HTC Coordinators, participants were aware of their leadership roles, one mentioned that:
“Having coordinators, who really know what they are doing is really important, because the whole team relies on them and needs that strong leadership.”(Government Policy Advisor, P1)


Most participants spoke about the critical leadership provided by the Senior Advisor of the DHHS, who was cited numerous times as possessing the necessary qualities to undertake a leading role in shaping HTV and securing the commitment of other leaders. One participant said that:
“The importance of strong leadership, that’s the ‘ [advisor’s name] effect’. It is the power, passion, energy and everything that is [advisor’s name] and what that translates to for motivation, workforce drive, stakeholder engagement. They were hugely powerful.”(HTC Evaluator, P14)


This quote flags the inter-relatedness between the HTV blocks of leadership, workforce and relationships. The importance of leadership was also linked to enabling the implementation of desired activities from Baugh Littlejohns and Wilson’s framework [35], with a participant expressing that:
“All of the leadership had a huge positive impact…it elevated the importance of the approach [systems thinking] which was fundamental to its success.”(HTC Officer, P20)


b. Implementation of desired activities

The delivery of coordinated and integrated interventions is a critical prevention system attribute. Accordingly, action was a key focus of HTV, yet surprisingly it was not stated in the HTV blocks. A participant identified this oversight:
“We struggled with where to put things like systems change actions. Where does that go? And the programs, social marketing and other strategies—the traditional stuff, where does that go? The interventions just didn’t fit neatly into a building block.”(Government Policy Advisor, P5)


Practitioners faced an implementation challenge caused by confusion regarding the role of projects within the systems approach, as they were required to deliver the Achievement Program and Healthy Lifestyle Programs. Some people asked, ‘where is the systems thinking in the Achievement Program?’ (HTC Team Leader, P21), while others thought of programs differently, for example a participant commented that:
“A program is a tool with capability, to leverage off the broader system, to lever broader conversations.”(HTC Team Leader, P29)


Participants’ confusion was an ongoing experience during their systems journey. Many of them spoke about a lack of clarity in defining what systems actions were and what they were not. One participant said that:
“I struggled to feel comfortable that I was actually working in systems. Everybody’s understanding was slightly different, there was always that sense of uncertainty about: am I actually doing what I need to be doing? Is this different to what I’ve previously done?”(HTC Officer, P27)


At the same time, other participants described how they had undertaken systemic actions such as: mapping systems using rich pictures; testing interventions using safe-to-fail experiments; and looking for leverage points using causal loop diagrams.

Confidence levels varied with systems concepts. The risk associated with uncertainty was practitioners’ temptation to resort to traditional methods during difficult times, such as the stress of funding cuts—demonstrating the links between Baugh Littlejohns and Wilson’s [51] implementation of actions part and the HTV blocks of workforce and resources.

c. Complex systems paradigm

HTV was embracing a complexity paradigm to improve the effectiveness of community-based health promotion, yet the characteristics of boundaries, leverage points, relationships and feedback, that are listed within the interpretative framework, are not in the HTV blocks. Even so, systems concepts were aspects of HTV and its contemporary approach to prevention. DHHS staff confirmed that systems sciences had been drawn upon in HTV’s design:
“There are a lot of different ways we talked about systems. We tried to encompass a lot of them and we used different ones in different ways. I don’t think that we had defined how we interpreted systems.”(Government Policy Advisor, P5)


Participants listed the systems tools that had influenced their work including the Cynefin Framework [55], the Foresight Obesity System Map [56] and the interrelationships-perspectives-boundaries (I.P.B.) framework [57]. Other health promotion theories had also been used, such as community development, place-based principles and settings approaches. A participant (P28) referred to experiencing ‘theory confusion’, as they did not understand the theories or how to use them (highlighting links between the complex systems paradigm and implementation of actions parts from Baugh Littlejohns and Wilson’s framework).
“There were too many theories. It wasn’t made clear what was actually at the core of what it was that we were doing.”(HTC Evaluator, P17)


The HTC workforce was reassured by the CEIPS that complex interventions, like HTV, required multiple theories to tackle public health challenges.

d. Information

The information part of the framework relates directly to the knowledge and data block of the HTV prevention system that includes the evaluation of HTV’s effectiveness. The CEIPS was established for this purpose and in 2013 developed a comprehensive evaluation plan. Unfortunately, it was only partially implemented so HTV was largely unevaluated. Many participants spoke about their disappointment, one said that:
“It has left the initiative and the systems approaches to prevention vulnerable. Which is unfair because we don’t know whether it worked or not.”(CEIPS staff, P19)


Participants were emotional about the lost opportunity to learn from this unique initiative. Another participant explained that:
“I feel desperately sad that we didn’t finish the information system for HTV.”(Government Policy Advisor, P4)


CEIPS started the difficult task of measuring prevention with the development of two tools: event logs to track practice to acquire local meaning and inform action (a link to Baugh Littlejohns and Wilson’s framework part of implementation of action for day-to-day practice); and a systems inventory to monitor change over time in the blocks. Intended surveys were never applied, regarding systems capacity, partnerships, networks and population health. Participants regretted that the evaluation tools had not been set-up from the start.

e. Collaborative capacity

The block regarding collaborative capacity in the HTV prevention system is relationships pertaining to partnerships for the purpose of achieving a common goal. The HTC workforce was effective at building partnerships, they had engaged with kindergartens, schools, workplaces and sporting clubs. This settings approach was established that allowed for cooperative events—demonstrating the interdependency between the HTV blocks of relationships and workforce as well as Baugh Littlejohns and Wilson’s framework part of implementation of actions. Collective impact was evolving concurrently in Victoria as well as in HTV and some HTC were broadening their stakeholder engagement for healthy eating to include food manufacturers, suppliers and retailers. This was underpinned by an awareness of local food systems. In addition to other agencies, practitioners had created internal relationships within councils, such as with departments of social planning, strategic planning, youth services, land-use and economic development. Practitioners were confident collaborators with an understanding of the necessity of partnerships:
“I had a pretty good handle on relationships and their importance. We were strongly connected. That permeates … across the system.”(HTC Evaluator, P18)


They knew that these connections were based on trust. These ties took time and effort to develop and expand.
“We did a lot of relationship mapping and trying to work out who we needed to engage with. Without someone to form those long-term partnerships—we could not do what we needed to do.”(HTC Coordinator, P7)


Participants needed to work with community health centres, as they had historically been the main provider of community-based health promotion. Several practitioners spoke about the challenge of partnering with community health as a barrier existed here:
“The community health people were put offside. We were disrespectful by not acknowledging that this was their space. We came in with a new approach that implied that their work had been bad. It created a big divide.”(HTC Team Leaders, P25)


With the establishment of HTCs the authority for community-based prevention shifted from community health. Community health staff were unsure of systems approaches and were reluctant to engage. Community health services held long-standing relationships, local knowledge and health promotion experience, and they were an important stakeholder for HTV. While they worked together, tension remained between them that hampered collaboration.

f. Health equity paradigm

The literature argues that an effective system needs to have equity embedded in its foundation in order to address social justice or human rights. However, participants did not speak about targeting vulnerable communities although equity was intended to be a guiding principle throughout HTV. It was not included in the blocks and perhaps not considered sufficiently, as one participant mentioned that:
“We talked about a population approach to bring everyone’s health up. That wasn’t thought through that well.”(CEIPS staff, P30)


g. Resources: financial

Participants unanimously described the resources allocated to HTV as being ‘huge’ and ‘unprecedented’ for prevention in Victoria. One interviewee commented that the funding was a:
“Significantly large amount of money not usually seen for health promotion.”(Government Senior Advisor, P3)


They were excited by this dedicated, yet flexible, funding arrangement between all levels of government. Prior to this, health promotion was considered to be under-funded and operating on the ‘smell of an oily rag’ (HTC Team Leader, P21). Participants felt that the sizeable sum could potentially holt the obesity epidemic. One participant explained that:
“This is the amount of money that I know is the right amount to make a difference”.(HTC Coordinator, P7)


Participants stressed the importance of the unallocated investment, that brought with it the opportunity to introduce systems thinking to health promotion and commence a new era of prevention. Those overseeing the funding best understood these advantageous circumstances and observed that:
“With new money you can create and innovate. You can do something different. In comparison, reforming existing dollars is much more difficult.”(Government Policy Advisor, P1)


In 2012 the resources were allocated to local governments and the HTC workforce was rapidly employed, signalling the commencement of HTV. The interdependence between the two HTV blocks resources and workforce was reinforced, following changes in state and federal governments when the incoming administrations withdrew funding, terminating the workforce and ending HTV in 2016. Participants discussed the implications of the reduced resources, not just that the duration of HTV had been reduced by several years, but also how it impacted on systems thinking, such as:
“The cuts were detrimental to systems practice. After each one, people went back to more traditional ways of working as they felt more confident doing it.”(HTC Officer, P22)
“Unreliable funding has been a big barrier to our systems work.”(HTC Evaluator, P18)


Many participants concluded that longer-term and more secure funding was needed to embed a systems approach within the HTV prevention system to achieve its full potential.

h. Resources: human

Human resources were the linchpin of both Baugh Littlejohns and Wilson’s framework and the HTV prevention system. Essentially the HTC workforce was needed to mobilize action and implement HTV’s systems thinking approach to community-based prevention (highlighting a critical dependency between workforce and the implementation of desired actions). Their recruitment was a priority to set the system in motion.
“Initially we worked on the workforce building block. Which was very much around pulling HTC together. We wrote a position description, for each position, and the job levels and where they would sit in the structure.”(Government Senior Advisor, P3)


The mix of positions was a deliberate consideration for the teams. They were intentionally structured of prescribed roles for HTV’s effectiveness. For example, a coordinator oversaw each HTC.
“Nobody ever put a health promotion person in a senior level, the way that we had in HTV. That’s why we created the coordinator’s role, to report to the CEO—be able to have strong influence. It is about leadership. You need to have leaders onboard otherwise you’re not going to see change.”(Government Senior Advisor, P28)


Evaluation and engagement roles were also defined, to ensure that these tasks were not forgotten in teams’ skill sets. It was anticipated that the initially inexperienced staff would rapidly develop systems knowledge and practice within a team environment. One participant commented:
“Being in a team, a sizeable team, was one of the most important parts of Healthy Together Communities.”(HTC Team Leader, P29)


It was referred to as the benefits of ‘group think’ (CEIPS staff, P19) to support the much-needed workforce development that was vital to shift implementation beyond traditional prevention methods. HTC practitioners described being offered an induction program, regular all-staff forums called Exchanges, specialist advice, an online forum, communities of practice and causal loop diagram workshops. However, this was insufficient for them to confidently acquire the practical-know how of applying systems thinking to health promotion, jeopardizing the theoretical basis of HTV. Participants were disappointed that:
“They [DHHS] charged us to do systems thinking without any real training or without any real support to do that.”(HTC Team Leader, P21)


Another threat to the workforce was the unanticipated staff turnover that resulted from two things. Firstly, some HTC practitioners did not enjoy the newfound systems practice and left HTV.
“Early on, sites lost staff. People signed up because they thought it was going to be different. The systems stuff challenged them, it did not suit their practice. They went to work somewhere else.”(HTC Coordinator, P10)


Secondly, the funding cuts prompted HTC practitioners to find more secure employment—displaying the causal link between the HTV blocks of resources and workforce. Participants noticed the impact of resignations:
“With every staff loss, went investments in terms of professional development and the experience they had gained in HTV. Momentum was also lost on strategies currently being implemented and the relationships built by the staff—that needed to be rebuilt whenever someone left.”(HTC Coordinator, P8)


#### 3.1.2. The Whole

Initially, the blocks were used to design HTV. However, once key blocks were established, less attention was paid to the whole system.
“The building blocks were pivotal in shaping the HTV roll-out. We veered away from it over time. In the beginning it was really important because it said what we needed to pay attention too for key investments in system building.”(Government Senior Advisor, P28)


The application of the prevention system was short lived when attention moved from the whole to the parts, prioritizing its sub-systems especially workforce and the implementation of activities. There seemed to be no system oversight or governance. Causal linkages were not explicitly discussed and there was no consideration of feedback loops.

### 3.2. System Map Findings

Figure 1a is a causal loop diagram of the HTV prevention system in practice, as described by the interview participants, including both building blocks from the initial HTV design as well as another identified via data analysis, as discussed above. It highlights the multiple links between the blocks in a highly connected and complex prevention system—the more interactions between the blocks, the more complex the system becomes. It also highlights the important role of the missing interventions block, akin to Baugh Littlejohns and Wilson’s framework part of implementation of desired actions. In Figure 1b three fundamental building blocks, knowledge and data, workforce and interventions to form the prevention system’s core feedback loop. They are connected by delayed positive causal links to make a reinforcing loop [58]. Reinforcing loops amplify change, either by spiralling growth (virtuous cycle) or decline (vicious cycle) [1]. The core loop was HTV’s driving force towards effective prevention via a systems approach, for example, the embedding of systems practice within the workforce was reliant upon this loop. Its neglect was a significant limitation of HTV that is described in the Discussion section.

## 4. Discussion

The use of the blocks for the establishment of the HTV prevention system was an exciting attempt to coordinate a macro community-based health promotion initiative. The new structure and complexity-informed approach to prevention was a HTV strength towards improving the effectiveness of large-scale interventions. It created a common language and shared understanding of the prevention system requirements. Other strong points included its: robust leadership that brought credibility to HTV; significant funding that enabled innovation; large workforce that set the system in motion; and the opportunity to reflect on the government’s return on the IHP investment [53]. While it is important to understand the strengths of the prevention system, the emphasis here will focus on the limitations and possible remedies for the purpose of advancing future health infrastructure initiatives and reforms.

The major limitation of the blocks model was that individual blocks were prioritized, rather than taking a whole-of system perspective of the prevention system. The macro system was considered for a short time, before attention shifted to the micro. For example, once the workforce was established there was little consideration of the entire prevention system. Consequently, the insights apparent via a holistic approach were not evident and the prevention system was not viewed as dynamic and highly connected—even though it was theorized as complex and adaptive [47]. Arguably, its core feedback loop went unnoticed.

### 4.1. Core Feedback Loop

The suggested core feedback loop includes three key blocks: workforce, the new interventions block and knowledge and data (see Figure 1b). 

#### 4.1.1. Workforce

With the workforce block, the HTV capacity-building strategies had created the necessary conditions for some of the HTC workforce to begin to fulfil their roles in a systems way [59]. Others felt that it was inadequate to establish confident systems thinkers. Coaching and peer support, in conjunction with tailored short courses that highlight implementation could have enhanced the systems thinking training [3,60]. More capacity ought to have been established as the workforce block could be influenced: as an endogenous variable it was mainly affected by other parts within the prevention system boundary (not by influences outside the system) and therefore it was able to be modified potentially more than it was [61]. This was a missed opportunity to create a culture of systems practice across blocks, as the workforce block was a key influencer to systems methods within the interventions block. Capacity building was a critical catalytic action in the prevention system, which is represented as a bold causal link in the core feedback loop in Figure 1b.

With high staff turnover, workforce capacity was exiting the prevention system. Resignations resulted from practitioners’ frustration transitioning to systems practice, this obstacle has been noticed elsewhere [21,62]. Other resignations were a consequence of the funding cuts and staff seeking more secure employment, creating an unstable workforce. This further hampered capacity, as the replacement employees were likely to be unskilled in system thinking and not offered professional development within the final three years of HTV.

#### 4.1.2. Interventions

The interventions block is the enabler of action, it is critical to the core feedback loop and to the success of the prevention system to improve public health [63]. This part forms a junction in the system as it is reliant upon inputs from all of the other parts to achieve its fullest potential, such as: dedicated funding for activities; knowledge for evidence-informed practice; leadership support for system methods; collaborative relationships; and a skilled workforce [64]. This interdependence within a complex system is an example of a part that cannot thrive without the whole.

The interventions block has two important functions, strategy design and activity delivery. Both can be the source of program failure via poor theory or inadequate implementation [65]. While the HTV systems approach to prevention was loosely informed by complexity frameworks, there was little direction how to operate other than to ‘adopt a systems approach’ [14,66]. In another multi-site systems-based initiative, the UK Healthy Towns Program, with little guidance for system practice, practitioners delivered programs (rather than engage in system interventions) as they did not know how to apply systems thinking [67]. Programs alone are insufficient to solve entrenched problems [30]. However, when they are utilised as a catalyst for change, they do have a role in systems approaches to prevention [68]. The use of programs as a disruptive event in a complex system was described by a limited number of interviewees. This may have been the intent for programs within HTV, but it was not widely understood by the HTC practitioners [69,70].

While complex interventions, like HTV, require multiple theories to tackle difficult problems, some HTC practitioners found them confusing, poorly justified and did not employ them and instead continued to use their past health promotion experiences [71]. Clear operating guidelines could have stimulated more systems practice and helped novice practitioners to define the norms and routines regarding what HTV was trying to establish, such as recommending methods to trial, for example systemic inquiry, soft systems methods or system dynamics [72,73]. While systems thinking was new to health promotion in Australia in 2012, at the time there were limited examples of it being applied elsewhere [74,75,76]. These experiences could have informed HTV’s systems approaches.

Beginner systems thinkers needed to know that a goal of complexity-informed prevention is to enhance the entire system as the collective behaviours create systemic outcomes [64,77]. Systems practice requires many interventions to generate multiple pathways of causality as change does not happen from one isolated point in the system, but is decentralized and the product of rippling processes occurring in concert over time [78,79]. One way to do this is via the synergy that comes from distributed and complementary interventions that influence a system’s self-organising ways [6,80], by working within it, to affect the contextual interactions that alter forces such as values, norms, practices, power, resources, policies, regulations and mental models [21,81,82]. Integrated multi-faceted interventions addressing these intrinsic leverage points, for example for healthy eating or physical activity, were not apparent in HTC micro level health promotion actions.

The interventions block was not identified in the explicit architecture of the HTV system, but of course it occurred and was part of the core feedback loop—its lack of explicit design was a problem. Other similar systems have accounted for implementation of activities [9,51,53], including the World Health Organization’s framework for strengthening health care systems, upon which the HTV prevention system was based [83]. Its omission caused limited critical thinking to occur, regarding the union of health and system sciences to reduce chronic disease. It seems to have been a mistake, rather than a deliberate act to move away from programs towards a wider range of actions, as the absence of the interventions block was identified by a Government staff member who was part of the team responsible for the development of HTV and confirmed that this also applied to systems change actions. The inexperienced workforce bore the brunt of the neglected application of theory to practice, requesting case studies and guidelines to better understand their roles. None the less, the new practice was well received by the workforce, even though challenges existed: learning while doing; acquiring systems language; conceiving projects as possible levers for systems change; and understanding what constitutes a systemic intervention versus what is not.

#### 4.1.3. Knowledge and Data

System evaluations seek to monitor changes in system’s drivers [84]. These types of measures were included in a comprehensive evaluation plan for HTV, with indicators regarding the blocks, the programs and system impacts [85]. Unfortunately, the plan was not implemented and the effectiveness of HTVs systems approach remains unknown [86]. Surprisingly, the lack of an evaluation may not have come from the knowledge and data block, but emerged from the surrounding blocks, such as leadership and resources. In complex networks no one factor can be singled out as the cause, as the outcome has arisen from the interaction of the parts over time, where the system’s behaviour is shaped by its structure [64,87].

#### 4.1.4. Change Takes Time

Prevention is a long-term prospect and delays are inherent in health promotion as the consequences of actions are rarely immediate [9]. The delay between the interventions block and the knowledge and data block comes as no surprise. For example, population-level health outcomes resulting from large-scale health promotion campaigns can occur decades after the intervention [58,88].

Another delay occurred in the link connecting the knowledge and data block and the workforce block. HTV needed to develop new monitoring tools and this lapse included the time to design and test the required technology [72]. As the tools were not finished, this delay became an obstruction in the core feedback loop. At this point, the core feedback loop was missing a connection (represented as a dotted line in Figure 1b), which may have been overcome if more time were available to HTV. This type of innovation is known to be inherently risky, time consuming and an obstacle to progress [11,72].

The delay between the workforce block and the interventions block was the time to employ and develop a large workforce. Staff required an establishment phase to acquire skills for the new practice [89,90]. A possible incorrect conclusion associated with this delay may have been that practitioners were able to utilize systems approaches sooner than they were.

While the discussion to this point has highlighted a range of issues relating to the core feedback loop, additional limitations of the prevention system existed including concerns regarding boundaries and systems oversight that are discussed below.

### 4.2. Boundary Issues

A system boundary is a conceptual line that separates a system from its environment by enclosing the parts of a system structure that are needed to generate the behavior of interest [91,92]. From the outset community health were marginalized, initially perceived as unimportant for the delivery of community-based prevention. However, in practice, they were drawn into the work of the HTC locally, contributing a community development approach, important connections and health promotion expertise [16]. With the benefit of hindsight, an interesting consideration is whether to have included community health services within the HTV prevention system boundary, rather than designing them outside of it. Shiell et al. (2018) discussed the need for greater consideration of the existing context in which social capital policies and programs are introduced and how to build from the strengths already located within a particular system [93]. Other authors similarly advise that effective interventions piggyback on the adaptive capacity of an existing system [57,94]. In the case of HTV, doing something completely different enabled greater innovation and locating HTC within local government provided access to their public health levers such as urban planning and economic development, endorsed by local governments’ legislated responsibilities of municipal public health and wellbeing plans [95,96]. However, the isolation from the existing health promotion infrastructure may have made HTV more vulnerable for discontinuation. While this is speculative, plans for future prevention system boundaries need to assess whether reforming an existing system or creating a new one is the best choice for its longevity and sustainability.

### 4.3. System Oversight

Like an orchestra without a conductor, there was no oversight of the HTV prevention system and no means of facilitating macro-level systems practice, such as redressing poorly functioning parts or absent connections. For example, staff retention planning across HTV may have strengthened the workforce block, in contrast to the governance that was focused on the individual HTC sites. The HTV prevention system may have been strengthened with a master plan and a coordinating mechanism, such as the Australian National Preventive Health Agency that existed from 2011 to 2014 to lead the national health promotion infrastructure [7]. Additionally, the mapping, safe-to-fail experiments and other systems practices that had been applied in HTC at the micro-level, had not been used at the macro-level, reinforcing that the blocks had not been viewed holistically by HTV [42].

### 4.4. Strengths and Limitations

The significance of this study is that it confirms the necessary ingredients of a prevention system and the importance of considering the whole system—the parts, causal linkages and feedback loops. The unique contribution that this research has made is the narrative of the intentionally built large-scale infrastructure for community-based prevention, the theoretical frameworks and testing them in the real-world application that was HTV. This adds to the limited research about the use of system sciences in health promotion, especially for the macro environment, that does not appear to be described elsewhere in the literature [14,15].

A feature of this research is its internal validity; credible knowledge was created through linking the interpretative framework to the research question via the data from a robust and representative sample that used verbatim quotes to enhance the study’s rigor [97]. A limitation of this research is that qualitative interviews are prone to bias, on behalf of both the participants and the researchers [98]. The researchers viewed themselves as actors inside the system that they were studying and therefore tried to mitigate their influences on the data by engaging a skilled interviewer, implementing interview protocols and scripts, and two researchers concurring on the coding.

## 5. Conclusions

In a large-scale effort to prevent chronic disease HTV applied systems theory to invigorate community-based health promotion by introducing a holistic approach led by local government. A highly connected system was created for a state-wide prevention platform. However, with no mechanism to oversee it, macro-level systems practice was never activated. The purpose of seeing the big picture and being able to redistribute resources to the needy parts of the system was unable to be performed. This was concerning as a new self-organizing and adaptive prevention system is unlikely to deliver effective and evaluated interventions without being nudged in the desired direction.

The HTV prevention system was not observed as a whole. It was treated as individual parts as the boundaries, causal relationships, feedback loops and emergent properties were not witnessed. The drawback of this reductionist perspective is that the insights provided by systems thinking were not harnessed. For example, the limited systems practice that emerged can be explained by the core feedback loop and the interactions between the blocks of workforce, interventions and knowledge and data. It was essential that this loop functioned to embed a systems thinking and acting workforce in HTV. However, with several delays, the loop was ultimately constrained by insufficient data. Regarding the parts of the loop, other systemic issues included:
More professional development was needed to establish routines and procedures for systems practice, within the highly adaptable workforce block.An interventions block was necessary within the prevention system, with operating guidelines to articulate and foster the norms and practices for community-based systems approaches to prevention.The time to develop evaluation tools of the knowledge and data block required longer than anticipated.


For inexperienced systems practitioners, these malfunctions increased their temptation to choose traditional approaches to prevention and to implement HTVs programs, rather than interact with the new and ambiguous methods. While the blocks provided a useful starting point for the HTV state-wide prevention system, this paper extends the concept by providing some direction for future research to examine how this framework could be adapted to create an effective macro-level architecture for complex health promotion interventions.

## Figures and Tables

**Figure 1 ijerph-18-01618-f001:**
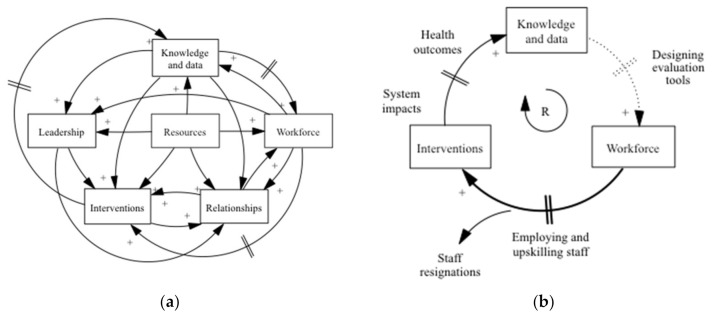
Healthy Together Victoria prevention system: in practice. (**a**) A causal loop diagram of the HTV prevention system building blocks. (**b**) The core feedback loop of the prevention system, excerpt from (**a**). Key: // a delay.

**Table 1 ijerph-18-01618-t001:** The organizational positions of the interview participants.

	Senior	Middle	Senior	Junior
	Management	Management	Practitioner	Practitioner
Healthy Together Communities	8	4	4	6
Department of Health and Human Services	3	3		
Centre of Excellence in Intervention and Prevention Science	2			1

**Table 2 ijerph-18-01618-t002:** Comparison between the elements of two prevention systems.

The Parts of Baugh Littlejohns and Wilson’s Framework, Strengthening Systems for Chronic Disease Prevention [51]:	The Building Blocks of the HTV Prevention System:
Leadership: political, cross-sector, multi-level, governance and accountability	leadership
Implementation of desired actions: multi-level, multi-faceted, knowledge-based, coordinated, integrated	
Complex system paradigm: boundaries, leverage points, relationships, feedback, diverse perspectives and contexts	
Information: surveillance, monitoring, research and evaluation	knowledge and data
Resources: financial	resources
Resources: human	workforce
Collaborative capacity: mindset, multi-sector and community	relationships
Health equity paradigm: social determinants of health, social justice and human rights	

Note: The Healthy Together Victoria building blocks are in small caps to distinguish them from the parts of Baugh Littlejohns and Wilson’s framework.

## Data Availability

Data is available on request from the corresponding author.

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
