# Peer review of "Building a Prevention System: Infrastructure to Strengthen Health Promotion Outcomes"

_ijerph, 2021, doi:10.3390/ijerph18041618_

Round 1

Reviewer 1 Report

Dear Authors,

I read your paper with interest. I believe your contribution is original since not many studies adopt system thinking with regard to health prevention. However, I believe some changes are required.

I believe that after the introduction you should add a theoretical background in which you briefly review the literature on prevention and system thinking, and you illustrate the Baugh Littlejohns and Wilson framework.

Subsections 1.1, 1.1.1 and 1.1.2 should be moved into the materials and method section.

Please change column order in table 1.

In the results section, please check the journal guide style for quotations.

I would split the result section into two main parts. One focusing on the interviews and one on the model (including the first part of the discussion).

Suggested readings

Homer, J. B., & Hirsch, S. M. (2006). System dynamics modeling for public health: Background and opportunities. American Journal of Public Health, 96(3), 452–458.

Bivona, E., & Noto, G. (2020). Fostering Collaborative Governance in Chronic Disease Management Programs: A Dynamic Performance Management Approach. In Enabling Collaborative Governance through Systems Modeling Methods (pp. 297-315). Springer, Cham.

Author Response

REVIEWER 1

Thank you for the feedback and the suggested readings, to assist with improving the paper.

Point 1.1 After the introduction you should add a theoretical background in which you briefly review the literature on prevention and system thinking, and you illustrate the Baugh Littlejohns and Wilson framework.

Response 1.1 Added rows 64-83, pg 2: ‘Since the early 2000s there have been ongoing requests in the literature calling for the use of systems thinking in health promotion, mainly to fill a void in the theory and practice of effectively tackling complex public health problems [1-3]. Systems thinking is drawn from the overlapping sciences of systems theory and complexity theory [4], where system dynamics and soft systems methodology are considered to be the relevant approaches to health promotion [3, 5, 6] that are currently being used in community-based prevention [7-11]. These methods shift the focus of health promotion activities away from a reductionist perspective to a holistic one [12], recognising that the essence of structure is not the parts themselves, but the relationships between the parts, their causal connections [13]. This is a fundamental shift in thinking for most health promotion practitioners moving from objectives-based planning and considering ‘Where do I implement a project?’, to problem-solving processes that view the complexity of an issues by asking ‘What needs to change to improve the situation?’ [14]. Systems thinking for prevention requires significant expertise, such as the reflecting, learning and adapting cycles of emergent strategy that facilitate system interactions by engaging communities, connecting agencies, coordinating assets, testing interventions, building consensus and aligning activities [15-18]. It involves taking risks and having the confidence to learn from them in dynamic situations to prevent obesity, family violence and other wicked problems [19].’

Point1.2 Subsections 1.1, 1.1.1 and 1.1.2 should be moved into the materials and method section.

Response 1.2 Subsections 1.1, 1.1.1 and 1.1.2 have been moved into the Materials and method section.

Point 1.3 Please change column order in table 1.

Response 1.3 In table 1, the column order has been changed.

Point 1.4 In the results section, please check the journal guide style for quotations.

Response 1.4 Sorry about that – the format changed on its way to you some how. The quotes have been reformatted.

Point 1.5 I would split the result section into two main parts. One focusing on the interviews and one on the model (including the first part of the discussion).

Response 1.5 The results section has been two sections: 3.1. Interview findings and 3.2. System map findings

Reviewer 2 Report

Thank you for your paper. It is interesting to read about the challenges implementing a systems perspective in prevention and promotion work. I feel that there are a few points that need attention if the paper is to be published.

Systems thinking is a concept with many variations. It is unclear from the introduction what you mean by systems thinking? Are you using complex adaptive systems? Or complexity theory as mentioned on page 2? This needs to be explained and why this choice has been made.

It would be worth discussing what system you are discussing. The different roles that are described do not mention that they include finance ministry, health ministry, or any ongoing evaluation. All three seem to be key in promoting health throughout a system. What sort of interventions were involved? Any new programs, working groups, campaigns? What was the main objective? (e.g. to increase healthy life years or embed health thinking key decision making processes) 

The Coding framework needs to be explained further. Why the Baugh Littlejohns and Wilson approach - writing because it is recent is not quite good enough. The HTV prevention system does not map onto this framework seamlessly so a little more reasoning is required to explain this choice. Also, what literature or other sources did you use to develop the codes? Generating codes as in Braun and Clarke is developed by more than just the data. 

The interview quotes are interesting. However, they are quite sparse and only one quote is used to explain some important points. Could your data be pushed further to explore the richness of the arguments that you are making. For example, in leadership you write that it was important to have a minister present at sites and coordinators who were aware of their role. Is this all the data says about leadership? What other practices and techniques are required? The HTV program ended in 2016. Is this an example of leadership failure or success?

In 3.1.3. Complex systems paradigm two quotes are used to indicate that system thinking was used. For example, 'DHHS staff confirmed that systems sciences had been drawn upon in HTV’s design: 

There are a lot of different ways we talked about systems. We tried to encompass a lot of them and we used different ones in different ways. I don’t think that we had defined how we interpreted systems. (Government Policy Advisor, P5)' 

It is a little unclear whether a systems thinking was used or systems tools (e.g. Foresight Obesity System Map). This could be explained in more detail. If it was  just the tools, how were they used? How does this fit with the idea proposed that a systems thinking was embedded throughout? 

Page 8 'challenge of partnering with community health as a barrier existed here'  

The quote indicates that the barrier was made by the new approach. Could be this explained further and how was it resolved? A single quote again for quote a substantial point seems inadequate.    

Page 9 - the cuts mentioned here suggest that the most important element to system thinking is funding? If so this needs to be explained further - why is money the most important element? How does this connect to leadership or collaboration? Why is systems reliant on new money rather than collaboration between funding streams/department budgets?

Page 11 - how was the causal loop digram made? Was it used developed during individual interviews or a systems workshop? Did the information provers have the opportunity to review and edit the diagram? 

Conclusion - did the HTV program have any material success? Did health improve? I feel that this could be explained more in the conclusion as it is hard to understand whether implementing a systems approach is beneficial for health and wellbeing of citizens in other regions and cities. 

Author Response

REVIEWER 2

Thank you for the feedback to assist with improving the paper.

Point 2.1 Systems thinking is a concept with many variations. It is unclear from the introduction what you mean by systems thinking? Are you using complex adaptive systems? Or complexity theory as mentioned on page 2? This needs to be explained and why this choice has been made.

Response 2.1 Add to line 63-66: Systems thinking is drawn from the overlapping sciences of systems theory and complexity theory [4], where system dynamics and soft systems methodology are considered to be the relevant approaches to health promotion [3, 5, 6]

Point 2.2 It would be worth discussing what system you are discussing. The different roles that are described do not mention that they include finance ministry, health ministry, or any ongoing evaluation. All three seem to be key in promoting health throughout a system. What sort of interventions were involved? Any new programs, working groups, campaigns? What was the main objective? (e.g. to increase healthy life years or embed health thinking key decision making processes) 

Response 2.2 To define the system added lines 54-60: ‘A prevention system includes the people, processes, activities, setting and structures, as well as the changing relationships between them, that work together to improve the health of a community [x]. Their core functions usually consist of policy, monitoring and surveillance, financing, research and evaluation, workforce, and program delivery [x] that together assist to recognise a system’s strengths and diagnose weak links to improve them [12].’

Examples are mentioned through out the paper such as those to prevent traffic injury, smoking, obesity and diabetes. Including the HTV goal, to reduce obesity and preventable chronic disease.

Point 2.3 The Coding framework needs to be explained further. Why the Baugh Littlejohns and Wilson approach - writing because it is recent is not quite good enough.

The HTV prevention system does not map onto this framework seamlessly so a little more reasoning is required to explain this choice. Also, what literature or other sources did you use to develop the codes? Generating codes as in Braun and Clarke is developed by more than just the data. 

Response 2.3 Baugh Littlejohns and Wilson (2019) framework was selected as it is contemporary and also because it was based on a systematic review and took a complexity / systems approach. (Perhaps that the HTV prevention system does not map onto the framework suggested that HTV was lacking rather than a problem with framework.)

The codes were largely based on the framework and also the interviewer’s key reflections. (That met the consensus of two researchers.)

While it is not mentioned in the paper, the research was based on constructivists epistemology and interpretivism theoretical perspective – to understand the sociocultural contexts and structural conditions, that enable the individual interview participant’s perspectives and multiple realities.

Point 2.4 The interview quotes are interesting. However, they are quite sparse and only one quote is used to explain some important points.

Could your data be pushed further to explore the richness of the arguments that you are making. For example, in leadership you write that it was important to have a minister present at sites and coordinators who were aware of their role. Is this all the data says about leadership? What other practices and techniques are required? The HTV program ended in 2016. Is this an example of leadership failure or success?

Response 2.4 In this paper selected quotes are used to illustrate certain aspects of the findings only. (Not presenting numerous quotes on the same topics.) This is to illustrate findings, signify different perspectives and show a richness of data. As described by Eldh, Arestedt and Bertero (2020), this is a valid use of quotes in qualitative research findings.

With respect to leadership, the paper describes it with respect to the research question in the context of establishing the HTV prevention system and systems thinking – not other aspects of leadership. The examples used are to highlight participants’ experiences with it, not if it was a success or failure, rather (as noted in the Discussion) ‘robust leadership that brought credibility to HTV’.

Ref: Quotations in qualitative studies: reflections on constituents, custom, and purpose. In International Journal of Qualitative Methods, 19: 1-6.

Point 2.5 In 3.1.3. Complex systems paradigm two quotes are used to indicate that system thinking was used. For example, 'DHHS staff confirmed that systems sciences had been drawn upon in HTV’s design: There are a lot of different ways we talked about systems. We tried to encompass a lot of them and we used different ones in different ways. I don’t think that we had defined how we interpreted systems. (Government Policy Advisor, P5)' It is a little unclear whether a systems thinking was used or systems tools (e.g. Foresight Obesity System Map). This could be explained in more detail. If it was just the tools, how were they used? How does this fit with the idea proposed that a systems thinking was embedded throughout? 

Response 2.5 The question pertains to the micro-level of systems practice, where ‘where health promoters are working in the system, achieving change with the local application of systems practices to tackle public health challenges’. While an overviews is described regarding a complex systems paradigm this question is beyond the scope of this papers (that focuses on the macro-level) and is described in another paper associated with this research, see: Bensberg, M. (2020). Developing a systems mindset in community-based prevention. Health Promotion Practice. doi: 10.1177/1524839919897266

Point 2.6 Page 8 'challenge of partnering with community health as a barrier existed here'. The quote indicates that the barrier was made by the new approach. Could be this explained further and how was it resolved? A single quote again for quote a substantial point seems inadequate.

Response 2.6 Added, at line 340: ‘With the establishment of HTCs the authority for community-based prevention shifted from community health. Community health staff were unsure of systems approaches and were reluctant to engage. Community health services held long standing relationships, local knowledge and health promotion experience, they were an important stakeholder for HTV. While they worked together, tension remained between them that hampered collaboration.’

Point 2.7 Page 9 - the cuts mentioned here suggest that the most important element to system thinking is funding? If so this needs to be explained further - why is money the most important element? How does this connect to leadership or collaboration? Why is systems reliant on new money rather than collaboration between funding streams/department budgets?

Response 2.7 Added to line 388, the HTV prevention system… to clarify that the HTV systems was reliant upon funding as it paid for the HTC workforce.

Indeed, systems thinking and collaboration are possible in an existing system – which is probably more sustainable, with ongoing (rather than short term) funding.

Point 2.8 Page 11 - how was the causal loop digram made? Was it used developed during individual interviews or a systems workshop? Did the information provers have the opportunity to review and edit the diagram? 

Response 2.8 The diagram was made from the themes of the interview data, as mentioned in line 186. (Not from a workshop.) No, the participants did not have the opportunity to edit the diagrams. This is a great idea – but it was not part of the process/protocol that was followed as described by Kim and Andersen (2012).

Point 2.9 Conclusion - did the HTV program have any material success? Did health improve? I feel that this could be explained more in the conclusion as it is hard to understand whether implementing a systems approach is beneficial for health and wellbeing of citizens in other regions and cities. 

Response 2.9: Great question. The answer is, that we don’t know the outcomes of HTV on wellbeing as (from line 574) ‘… These types of measures were included in a comprehensive evaluation plan for HTV, with indicators regarding the blocks, the programs and system impacts [85]. Unfortunately, the plan was not implemented and the effectiveness of HTVs systems approach remains unknown.’

Reviewer 3 Report

Please see my comments attached below for the authors consideration.

All the best.

Author Response

REVIEWER 3

Thank you for the feedback to assist with improving the paper.

Point 3.1 Abstract: For your consideration a sentence(s) to set the context of systems thinking and health promotion research and practice would be a good addition to the Abstract as a topic sentence(s). A link to Australia and set the geographical context I believe would also be helpful in the Abstract and also please include in the main paper –Introduction.

Response 3.1 Added to the Abstract: Prevention systems improve the performance of health promotion interventions.

Added rows 64-83, pg 2: ‘Since the early 2000s there have been ongoing requests in the literature calling for the use of systems thinking in health promotion, mainly to fill a void in the theory and practice of effectively tackling complex public health problems [1-3]. Systems thinking is drawn from the overlapping sciences of systems theory and complexity theory [4], where system dynamics and soft systems methodology are considered to be the relevant approaches to health promotion [3, 5, 6] that are currently being used in community-based prevention [7-11]. These methods shift the focus of health promotion activities away from a reductionist perspective to a holistic one [12], recognising that the essence of structure is not the parts themselves, but the relationships between the parts, their causal connections [13]. This is a fundamental shift in thinking for most health promotion practitioners moving from objectives-based planning and considering ‘Where do I implement a project?’, to problem-solving processes that view the complexity of an issues by asking ‘What needs to change to improve the situation?’ [14]. Systems thinking for prevention requires significant expertise, such as the reflecting, learning and adapting cycles of emergent strategy that facilitate system interactions by engaging communities, connecting agencies, coordinating assets, testing interventions, building consensus and aligning activities [15-18]. It involves taking risks and having the confidence to learn from them in dynamic situations to prevent obesity, family violence and other wicked problems [19].’

Line 87 added .. ‘Within Australia’.

Line 9 added: ‘the Australian state government initiative’.

Point 3.2 Introduction, para 2 remove the word accident and use road traffic injury or crash as the injury prevention space prefers not to use the word ‘accident’.

Response 3.2 Line 49: changed ‘accident’ to ‘injury’.

Point 3.3 Para 3 – delete the methods description i.e.‘ Semi-structured interviews (n=31).... and the results summary text out of the introduction section and add the geographic and or the context of Victoria being a state in Australia into section 1.1 Healthy Together Victoria

Response 3.3 Deleted the suggested text. And see response 3.1.

Point 3.4 Methods section Did the authors use the Consolidated Criteria for Reporting Qualitative Research (COREQ) to guide the interview design to ensure comprehensive reporting and prevent bias, increase trustworthiness etc (Tong, Sainsbury & Craig, 2007)?

Response 3.4 Yes, the authors did use the COREQ (Tong, Sainsbury and Craig, 2007) for reporting this qualitative research. The guidelines for authors by Malterud (2001) was also used. (Qualitative research: standards, challenges and guidelines in The Lancet, vol 358, 2001.)

Point 3.5 Page 3 paragraph 2 can you discuss some of the pragmatics of the thematic analysis in a little more detail e.g. the themes seem to be mostly aligned with Baugh Littlejohns and Wilsons framework and I am not sure ‘themes have emerged and been refined’ as such? how was consensus reached for the themes in the table to be used for the reporting of the results, was for example any disagreement, did more than one researcher undertake the analysis?(this is noted as a strength on page 15of 19 but I believe it needs to be better outlined in the methods section). How was the rigour and trustworthiness of the data maintained? Perhaps some additional information on the data analysis would be of value to the reader here e.g.was a reflective journal kept, especially as part of systems practice is being iterative and reflexive.

Response 3.5 On page 3, lines 160-163 added: ‘The analysis was undertaken by one researcher and discussed with a second to reach consensus on coding and themes. A third researcher had been invited to resolve any analysis discrepancies but this was not needed.’

Line 155-156 added: After each interview, the interviewer documented their key reflections.

Line 165-166 added: The themes were considered along side the interviewer’s key reflections.

Point 3.6 Were any demographics collected on the Interviewees? Is this available to include in the results section?

Response 3.6 Table 1 was added on page 3 showing the organisational positions of the interview participants

Point 3.7 Results: The first sentence could include a note that the that the verbatim quotes from participants were used to provide thick, rich description (see Lindseth & Norberg; 2004). Names and other identifying factors were removed for reporting, could also be a good addition.

Response 3.7 Added to the study design, in lines 151-153: ‘The participant’s name was not recorded or included in the transcript. Other factors that may have identified a participant were removed from the transcript. Two participants took up the offer to audit their interview transcripts.’

Point 3.8 Page 5 –paragraph 4 the pronoun ‘she’ is used to describe a Senior Advisor, I wonder if this is potentially identifying and maybe could be removed and replaced with a non-gender specific label.

Response 3.8 In line 212 , removed ‘she’ and replaced it with ‘who was’. In line 218 replace ‘she’ with ‘they were’.

Point 3.9 Page 10 –check if the word ‘complained’ is the best word for the paper –it seems a judgement call and maybe should just read participants ‘highlighted’ or ‘felt’.

Response 3.9 Changed ‘complained’ to ‘were disappointed’ in line 409

Point 3.10 Discussion: There are some good points posed by the authors McGill et al., 2019 in their paper (see Figure 4) Levers of Change, specifically around the evaluation of a systems approach and I wonder if they are worth citing in your Discussion in section 4.1.3.see: Egan M, et al. NIHR SPHR Guidance on Systems Approaches to Local Public Health Evaluation. Part 2: What to consider when planning a systems evaluation. London: National Institute for Health Research School for Public Health Research; 2019.

Response 3.10 Thank you for the recommended reference. I am familiar with the document. In an earlier version of the paper evaluation was described – but the paper was getting too long and not providing enough detail on developmental evaluation – so it was removed. It was beyond the primary scope of this paper

Point 3.11 Conclusion: Well written but quite long, consider moving some text to the Discussion e.g. para 2 of the Conclusion could be shortened and just focus on the issues and the recommendations you have outlined to action the issues?

Response 3.11 The Conclusion is about 350 words, in a paper of about 7 000 words – is that too long? As other reviewer’s did not comments about the conclusion, no changes have been made to it.

Point 3.12 General considerations: Conclusion –page 16 approx para 4 -check if you need to add a ‘d’ to ‘For inexperience systems practitioners...’ and it should read ‘For inexperienced systems practitioners...’

Response 3.12 ‘d’ added to inexperience to line 645 (when the change was made)

Point 3.13 Review the formatting of the references the Endnote seems to be using abbreviations incorrectly (you need a comma after the organisation name in Endnote) see ref 28 and 75 as an example

Response 3.13 All referenced have been reformatted.

Round 2

Reviewer 2 Report

Accept paper.